# Systematic Review on Diversity and Distribution of *Anopheles* Species in Gabon: A Fresh Look at the Potential Malaria Vectors and Perspectives

**DOI:** 10.3390/pathogens11060668

**Published:** 2022-06-09

**Authors:** Neil Michel Longo-Pendy, Larson Boundenga, Pyazzi Obame Ondo Kutomy, Clark Mbou-Boutambe, Boris Makanga, Nancy Moukodoum, Judicaël Obame-Nkoghe, Patrice Nzassi Makouloutou, Franck Mounioko, Rodolphe Akone-Ella, Lynda Chancelya Nkoghe-Nkoghe, Marc Flaubert Ngangue Salamba, Jean Bernard Lekana-Douki, Pierre Kengne

**Affiliations:** 1Unit of Vector Systems Ecology of Interdisciplinary Centre for Medical Research of Franceville (CIRMF), Franceville BP 769, Gabon; longo2michel@gmail.com (N.M.L.-P.); judicael.obame@live.fr (J.O.-N.); fmounioko@yahoo.fr (F.M.); ousmaneakone@gmail.com (R.A.-E.); lyndankoghe@gmail.com (L.C.N.-N.); ngangmarcfaub@yahoo.fr (M.F.N.S.); pierre.kengne@ird.fr (P.K.); 2Unit of Wildlife Parasites and Neglected Parasitosis, Parasitology Department of Interdisciplinary Centre for Medical Research of Franceville (CIRMF), Franceville BP 769, Gabon; clarkmbou@gmail.com (C.M.-B.); nancydiamella@gmail.com (N.M.); patmak741@gmail.com (P.N.M.); 3Department of Anthropology, Durham University, South Road, Durham DH1 3LE, UK; 4National Malaria Control Program, Health Ministry (PNLP), Libreville BP 50, Gabon; piazzy09@yahoo.fr; 5Animals Biology Department of Sciences Faculty, Cheikh Anta Diop University (UCAD), Dakar BP 5005, Senegal; 6Research Institute for Tropical Ecology (IRET/CENAREST), Libreville BP 13354, Gabon; makanga.boris@gmail.com; 7Unit of Evolution, Epidemiology and Parasite Resistance, Parasitology Department of Interdisciplinary Centre for Medical Research of Franceville (CIRMF), Franceville BP 769, Gabon; lekana_jb@yahoo.fr; 8Biology Department, Masuku University of Sciences and Technic (USTM), Franceville BP 943, Gabon; 9Department of Parasitology, Health Science University (USS), Owendo, Libreville BP 4009, Gabon; 10MIVEGEC, IRD, CNRS, University of Montpellier, Montpellier 911 Avenue Agropolis, BP 64501, 34394 Montpellier, France

**Keywords:** *Anopheles* species, diversity, spatial distribution, sylvatic, rural, urban, Gabon

## Abstract

Gabon is located in the malaria hyper-endemic zone, where data concerning malaria vector distribution remains fragmentary, making it difficult to implement an effective vector control strategy. Thus, it becomes crucial and urgent to undertake entomological surveys that will allow a better mapping of the *Anopheles* species present in Gabon. In this review, we examined different articles dealing with *Anopheles* in Gabon from ProQuest, Web of Science, PubMed, and Google scholar databases. After applying the eligibility criteria to 7543 articles collected from four databases, 42 studies were included that covered a 91-year period of study. The review revealed a wide diversity of *Anopheles* species in Gabon with a heterogeneous distribution. Indeed, our review revealed the presence of 41 *Anopheles* species, of which the most abundant were members of the Gambiae and Nili complexes and those of the Funestus and Moucheti groups. However, our review also revealed that the major and minor vectors of malaria in Gabon are present in both sylvatic, rural, and urban environments. The observation of human malaria vectors in sylvatic environments raises the question of the role that the sylvatic environment may play in maintaining malaria transmission in rural and urban areas. Ultimately, it appears that knowledge of biodiversity and spatial distribution of *Anopheles* mosquitoes is fragmentary in Gabon, suggesting that additional studies are necessary to complete and update these entomological data, which are useful for the implementation of vector control strategies.

## 1. Introduction

Despite several decades of control efforts, malaria remains one of the major public health problems in sub-Saharan African countries [1,2,3]. Six species (*Plasmodium falciparum*, *Plasmodium vivax*, *Plasmodium malariae*, *Plasmodium ovale wallekeri*, *Plasmodium ovale curtesi*, and *Plasmodium knowlesi*) are known to cause this disease in humans worldwide [4,5,6]. According to the WHO, in 2020, approximately 241 million clinical cases and 627,000 deaths worldwide were attributable to malaria [2]. Sub-Saharan Africa accounted for 95% of malaria cases and 96% of malaria deaths [2]. Among the victims of malaria, children under the age of 5 years remain the most vulnerable population [7,8]. Indeed, they account for 80% of all malaria deaths in Africa [2]. Thus, malaria is proving to be an impediment to economic prosperity in many tropical countries, particularly in Africa [9].

The circulation of malaria in populations is ensured by mosquitoes belonging to the genus *Anopheles*. Approximately 41 species are known to be major vectors of this disease, such as species of the Gambiae complex [10,11], Nili [12], or those of the Funestus [13], and Moucheti groups [14]. However, there is a wide diversity of *Anopheles* species that are capable of transmitting the disease [3,15]. Currently, vector control is an essential and effective method recommended by the WHO to fight and eliminate malaria. The method relies on the use of long-lasting insecticide-treated mosquito nets (LLINs) and indoor domiciliary spraying (IRS) for adult mosquitoes. According to WHO guidelines for malaria in 2021, larviciding is considered a supplementary measure implemented alongside ITNs or IRS for urban areas where breeding sites are relatively few, fixed, and findable in relation to houses [16]. However, the effectiveness of this method depends on a better knowledge of the bioecology of the vectors in a given region or locality [17,18]. It is, therefore, crucial and urgent to undertake entomological surveys that will allow a better mapping of the *Anopheles* species present in a given region and to monitor the responses of *Anopheles* communities to different interventions and climate change [19,20].

Gabon is located in the malaria hyper-endemic zone [7,21,22], but the distribution of malaria vectors in this country or their relationship to malaria transmission remains fragmentary [23]. The studies carried out covered only certain areas of the country [24,25,26,27], and some of them are more than 60 years old [28,29,30,31], indicating that these data are not representative of the anopheline diversity that the Gabonese ecosystem may harbor. Recent studies in sylvatic environments have harbored a wide diversity of *Anopheles* species, among which some are known to be vectors of human malaria in Gabon (*An. moucheti*, *An. gambiae*, *An. Funestus*, and *An. nili*) [25,32,33,34,35]. This raises the question of the role played by forest ecosystems in maintaining malaria transmission in urban areas. It, therefore, appears very interesting to make available to the scientific community a summary document showing the global and updated spatial distribution of *Anopheles* species described in Gabon. With this in mind, we compile in the systematic review all the published studies carried out in Gabon on *Anopheles* mosquitoes in different ecosystems, highlight the role that forest ecosystems could play, and discuss research perspectives with a view to providing some further information that could be used for the development of a national vector control strategy.

## 2. Materials and Methods

### 2.1. Search Strategy

This study retrieved articles from five science databases, namely, ProQuest, Web of Science, PubMed, and Google scholar. A systematic review was performed using a predefined protocol based on the preferred reporting items for systematic reviews and meta-analyses (PRISMA) [36,37]. The searching process utilized the following three keywords: “*Anopheles*”, “AND”, and “Gabon”. In order to reduce the risk of bias from the number of articles obtained, the researchers researches carried out made conducted disbursements in all databases using the same keywords and on the same day.

### 2.2. Eligibility Criteria

To establish an updated spatial distribution of *Anopheles* mosquitoes in Gabon, we included all studies that considered both adults and larvae *Anopheles* mosquitoes collected in Gabon. Thus, this review included studies that considered all field studies in different environments whether sylvatic, rural, or urban. We also considered all other scientific publications (opinions, scientific reports and perspectives, and duplicate records) that did not focus primarily on *Anopheles* but on other Diptera while mentioning an *Anopheles* species in their results. However, we excluded articles for which the full version was not available, those mentioning the results of other articles and studies with no relevance to our topic.

### 2.3. Study Selection

The articles’ eligibility was determined from each title, abstract, and full text by two reviewers (LA and NLP). They also independently screened the articles for inclusion and extracted data on general information. To resolve any disagreements about the inclusion or exclusion of an article, or an abstract of an article, that arose during the course of their individual reviews, the researchers met regularly to discuss the issue. Some disagreements were resolved through involvement of another person (PK).

### 2.4. Data Extraction and Analysis

After resolving the differences in data extraction or interpretation through consensual discussions based on the inclusion and exclusion criteria mentioned above, the final papers were selected. Data from eligible studies included the following: mosquitoes of *Anopheles* genus, species, year of study, study area where study was conducted, habitat characteristics, country, and distribution of *Anopheles*. The results of our analysis were classified according to the purpose of our study. Throughout our analysis, the results in one locality by one author were compared with the results of other authors working in same locality. Thus, the data were retained in relation to *Anopheles* diversity, spatial distribution, and information as a vector of malaria. All variables for which we extracted data were the following: *Anopheles* species, habitat (urban, rural, and sylvatic), status (major or minor malaria vector), and *Plasmodium* infestation. To reduce the bias risk, we selected all articles, abstracts, and reports in French and English. In addition, to assess bias risk, we used SYRCLE’s tool (Systematic Review Center for Laboratory Animal Experimentation’s) for animal studies [38,39]. It is a perform well tool that includes ten domains with six types of bias.

All statistical analyses and graphs were performed on R software version 4.0.2. In order to highlight the logical-mathematical relationships, in particular the similarities and differences in composition of *Anopheles* species between the three environments (urban, rural, and sylvatic), a Venn diagram was made using the package “Venn Diagram” [40]. Moreover, all bar charts were made using the Ggplot2 package [41].

## 3. Results

### 3.1. Search Results

After analyzing the different databases, we collected 7543 articles extracted mainly from the four search engines Google Scholar (5062 articles), Web of Sciences (50), PubMed (46), and ProQuest (2385 articles) (Figure 1). After removing duplicates and screening titles and abstracts, 56 records were included for full-text assessment. Fifteen (15) articles were excluded with reasons (for more information, see Figure 1), while 42 articles [24,25,26,27,28,29,30,31,32,33,34,35,42,43,44,45,46,47,48,49,50,51,52,53,54,55,56,57,58,59,60,61,62,63,64,65,66,67,68,69,70,71] fully met the inclusion criteria.

### 3.2. Study Characteristics

The included papers consisted of studies conducted in three distinct environments, specifically twenty-two (22) were urban, eleven (11) rural, seven (7) forest, and two (2) studies conducted in both rural and urban settings. In total, forty (40) *Anopheles* species were reported in the 42 studies, of which thirty-eight (38) were known [72] and two (2) undetermined. Among these species, some of the following were reported in at least two studies: *Anopheles gambiae s.s*. (17), *Anopheles moucheti moucheti* (12), *Anopheles funestus* (10), *Anopheles coustani* (8), *Anopheles paludis* (7), *Anopheles nili s.s.* (7), *Anopheles marshallii* (7), *Anopheles coluzzii* (6), *Anopheles hancocki* (4), *Anopheles gabonensis* (4), *Anopheles obscurus* (4), *Anopheles vinckei* (3), *Anopheles rufipes* (3), *Anopheles melas* (3), *Anopheles carnevalei* (3), *Anopheles theileri* (3), *Anopheles ziemanni* (2), *Anopheles wellcomei* (2), *Anopheles tenebrosus* (2), *Anopheles squamosus* (2), *Anopheles pharoensis* (2), *Anopheles mauritianus* (2), *Anopheles maculipalpis* (2), *Anopheles jebudensis* (3), *Anopheles hargreavesi* (2) and *Anopheles cinctus* (2). However, some of them were only studied by a single study, notably *Anopheles arabiensis*, *Anopheles demeillioni*, *Anopheles implexus*, *Anopheles moucheti nigeriensis*, *Anopheles pretoriensis*, *Anopheles rodhesiensis*, *Anopheles fontenillei, Anopheles smithii s.l., Anopheles faini, Anopheles schwetzi,* and *Anopheles eouzani* (Figure 2, Table 1 and Appendix A).

### 3.3. Risk of Bias Assessment

Except for ten studies [28,29,30,35,43,44,53,59,62,66], which were at low risk for data acquisition, all 31/41 studies reviewed were at a high probability of reliable data generation in diversity and distribution of *Anopheles* species and sites of studies. Concerning sites where studies took place, only one [43] had an unclear risk. The studies presented a detailed and consistent reporting of all outcomes pre-specified on *Anopheles* species. All 42 studies had a low risk of attrition and reporting bias. However, we believe that the absence of certain data, such as the number of *Anopheles* mosquitoes collected in some of the studies analyzed for this review, is unlikely to influence the current findings.

### 3.4. Spatial Distribution of Anopheles Species in Gabon

The census of species reported at the different sites allowed us to produce a spatial distribution map of *Anopheles* species in Gabon (Figure 3). This spatial visualization allowed us to highlight study areas and species of *Anopheles* identified in Gabon in different environments (urban, rural, and sylvatic). Concerning the urban environment, the data used to produce this map (Figure 3A) came from 23 articles and concerned only the provincial capitals, such as Libreville, Franceville, and Port-Gentil. In urban areas, twenty-three (23) *Anopheles* species were reported. In rural areas (Figure 3B), the map was generated from data collected in 12 articles whose study sites concerned several villages (Dienga, Benguia, Coco Beach, Mitzic, Mekambo, Booué, Fernan Vaz, Omboué, Mourindi, Mayumba, Ndjole, and Bakoumba) away from coastal areas. In rural areas, seventeen (18) *Anopheles* species were reported, such as *An. gambiae s.s.*, *An. Paludis,* and *An. cinctus*. In the forest environment (Figure 3C), only six studies were able to identify and distribute *Anopheles* species, but only in two sites, namely, the Lopé and Lékédi parks. Twenty-four (24) *Anopheles* species were reported in this environment, such as *An. carnevalei, An. Jebudensis,* and *An. maculipalpis*, despite the small number of studies conducted in this environment.

However, the Venn diagram reveals that among the *Anopheles* species found in Gabon, some are specific to a particular environment (*An. melas* and *An. rufipes* in urban coastal areas; *An. fontenillei* and *An. vinckei* in sylvatic areas; *An. moucheti nigeriensis* and *An. wellcomei* in rural areas). On the other hand, other species are more generalist because they have been found in several environments (*An. coluzzii*; *An. moucheti*; *An. gambiae*; *An. funestus*; *An. nili s.s.*; *An. paludis*) (for more details, see Figure 4).

### 3.5. Abundance of Anopheles Species Reported

Analysis of the cumulative abundance of *Anopheles* species between the years 1931 and 2022 shows that each species was observed at least once in Gabon (Figure 5A). Thus, among the 32,633 specimens of *Anopheles* spp. recorded in the 42 articles covering a period of 91 years (from 1931 to 2022), the most abundant species are members of the *An. gambiae* and *An. nili* complexes and of the *An. funestus* and *An. moucheti* groups, representing 33%, 9%, 32%, and 18%, respectively, of all *Anopheles* recorded (Figure 5B).

### 3.6. Reservoir Role of the Sylvatic Environment for Malaria Vectors

The results of our review reveal that among all the species recorded in these different studies, some have a strict anthropophilic preference, such as *An. gambiae s.s*. and *An. funestus*, and a strict primatophilic preference, such as *An. vinckei*. However, species such as *An. marshallii* and *An. moucheti* have been found infected with both human and primate *Plasmodium*.

## 4. Discussion

Malaria is a global burden that causes thousands of deaths every year, with the heaviest burden still being paid by African countries, particularly those in Sub-Saharan Africa. Thus, to fight this disease, many countries have put in place malaria control strategies. However, the implementation of a better control strategy requires good mapping of the disease and of the ecology of the potential vectors and reservoirs of this disease.

In the present study, we reviewed the literature to provide a global mapping of the diversity and spatial distribution of *Anopheles* in Gabon. We also examined the role of different ecosystems (urban, rural, and sylvatic) in the distribution of malaria vectors in mammals, including humans. To our knowledge, this could be the first systematic review assessing the distribution and diversity of *Anopheles* in the different environments mentioned above in Central Africa in order to highlight the lack of information needed in Gabon for the implementation of a reliable malaria vector control strategy. Understanding the ecology of the mosquito is important for implementation. The studies included in our review were of good quality because most had a small margin of error in species identification. However, a small number of the included publications (n = 9) did not provide clear information on the number of individuals observed or identified by species and site. We believe that this may not have a real influence on the results obtained in our review.

Regarding the diversity of species belonging to the genus *Anopheles* reported in Gabon so far, the analysis of the available literature data shows that only a few articles mention malaria vectors. All the included studies covered a period of 91 years (1931, the year of the oldest publication, and 2022, the year of writing this review). In total, at least 40 *Anopheles* species have been recorded in Gabon in different ecosystems by a limited number of publications during nearly a century of study by different teams. Our results revealed that the species richness (diversity of species) was more important in sylvatic areas (forest environments) with at least twenty-eight (28) species (including two new species never recorded, *An. gabonensis* [49] and *An. fontenillei* [34]) for only a few studied sites. However, we believe that the number of species in the sylvatic areas could be even greater, as the studies by Makanga et al. (2016) and Obame-Nkoghe et al. (2017) highlight the presence of indeterminate species based on current morphological criteria [25,71]. These data could suggest the existence of one or more other species in the forest environment. However, this remains to be demonstrated through further investigation.

Surprisingly, we observed that the species richness (diversity) of *Anopheles* was similar between the sylvatic and urban environments. This does not corroborate with previous studies that suggest a lower species diversity of *Anopheles* in anthropized environments [73,74]. This result could be explained by the fact that the studies that provided the greatest number of species in urban areas date back to the colonial period between the 1930s and 1970s [28,29,30,31], during which the level of urbanization of cities was not the same as today [75,76]. In contrast, very recent studies in Libreville [47,51,52,70] and in Port-Gentil [26] and Mouila [45], report only 2–3 species collected. Thus, we believe that additional studies would be necessary for a better assessment of the impact of anthropization on the diversity of *Anopheles* species found in this environment. The loss of diversity observed in urban and rural habitats compared to the sylvatic environment could be explained by the fact that anthropogenic modifications were undertaken in these habitats, such as forest destruction, urbanization, and agriculture, which could have had a considerable impact on the dynamics of mosquito vectors and *Anopheles* in particular [77,78]. Moreover, the contamination of breeding sites of species by chemical or organic substances linked to urbanization [73] could contribute to the decrease or even disappearance of preferential breeding sites for certain species, as described for *An. gambiae s.s*. [79].

The results of the spatial distribution of these species and the Venn diagram have allowed us to highlight two groups of species as follows: those that we call “generalists” found in at least two of the three environments (*An. coluzzii*, *An. gambiae s.s.*, *An. funestus*, *An. nili s.s*, *An. moucheti moucheti*, *An. paludis*, *An. coustani*…); those qualified as “specialists”, i.e., those that are specific to a particular environment (*An. carnevalei* and *An. fontenillei* in the sylvatic environment, *An. moucheti nigeriensis* in the rural environment, *An. melas* and *An. rufipes* in the urban environment in Gabon). The difference observed between these two groups could be explained by the adaptive potential of the different species to disturbed environments. Indeed, some species are known for their ability to colonize several environments different from their original habitat, such as some species of the *An. gambiae* complex (*An. coluzzii* and *An. gambiae*) [51,80,81]. In particular, the generalist group includes species such as *An. gambiae s.s.*, *An. coluzzii*, *An. funestus*, *An. Moucheti*, and *An. nili*, which are known to be major malaria vectors in sub-Saharan Africa [12,14,82,83] and, in particular, in Gabon [35]. Furthermore, our analysis reveals a worrying situation represented by the fact that members of this group were the most abundant in all studies conducted so far (*gambiae* complex 33%, *funestus* group 32%, *moucheti* group 18%, *nili* complex 9%, and the other species represented only 8%). This situation could constitute a brake or obstacle in all malaria vector control strategies insofar as forest environments could constitute refuges and/or reservoirs for these species ensuring the transmission of malaria to humans. On the other hand, in the absence of these major species due to climatic factors such as the season [84,85], the secondary species of this group (*An. paludis, An. marshallii, An. Squamosus,* and *An. coustani*) could take over the transmission of malaria in the populations living in these different environments. This would have the potential consequence of perpetuating transmission throughout the year [86,87]. Moreover, the presence of human malaria vectors and other zooanthropophilic *Anopheles* species in this group of generalists could raise the issue of interspecies transmission [33,48], which would result from increased contact between human populations and these vectors as a result of anthropogenic activities such as the increase in logging camps, agriculture, hunting, gathering, and ecotourism activities. This observation highlights the importance of a “One Health” approach in the effective control of infectious vector-borne diseases with zoonotic potential.

These different study sites are not representative of Gabon. Most of the studies concerning the taxonomy, biology, and genetics of *Anopheles* fauna in Africa, in general, and Gabon especially, have focused primarily on anthropophilic species, and the characterization of zoophilic species is, therefore, far from complete. So then, there is a significant scientific gap when it comes to vectors of pathogens circulating in sylvatic wildlife [49]. However, data available about vectors bring some information on *Anopheles’* ecology and distribution. It is true that very few things about the biting behavior of sylvan *Anopheles* and about the vertebrate hosts constitute their preferred source of blood [33]. However, recent studies indicate that, concerning their blood meals, many species are opportunistic (e.g., *An. vinckei* and *An. gabonensis*). Such a propensity to bite a wide range of hosts is probably an adaptive trait in response to temporal fluctuations of host diversity and density in forest environments. This feature could enhance the possibility for cross-species transfer of parasites and could explain the parasite’s propensity to infect different host species [32]. Certain species frequent a number of environments (forests, savannas, and urban areas). *An. gambiae*, *An. Nili,* and *An. Moucheti,* which are vectors of human malaria, were observed in the forest [25], which means that their distribution is not only limited to urban areas.

## 5. Conclusions and Perspective

The presence of a large diversity of *Anopheles* species hosted by different ecosystems encountered in Gabon (sylvatic, rural, and urban) offers interesting research perspectives in the field of ecology of malaria transmission. Knowing that the implementation of a best vector control strategy requires a better understanding of the mapping of vector species (characterization of the bio-ecology of vectors, factors determining their distribution, and phenotypes/genotypes of resistance to insecticide molecules on the market) [88,89,90], the lack of this data would constitute a real gap for the implementation of an effective anti-malaria strategy in Gabon.

At the end of this study, we notice that Gabon presents an (*i*) lack of current entomological data on malaria in almost 2/3 of its territory. This observation could be due to the fact that the main studies carried out were in easily accessible areas to the detriment of the interior of the country; (*ii*) the absence of updated data in several regions that have undergone anthropic modifications; (*iii*) insufficient data on entomological indices of malaria transmission, susceptibility of vectors to insecticides, and mechanisms of insecticide resistance. As a direct consequence, the vector control strategy implemented in Gabon is limited. Thus, the gaps observed show that scientific researchers must direct their research towards the establishment of a better national strategy for the control of malaria vectors, which would take into account all the compartments involved in the dynamics of *Anopheles* mosquitoes in Gabon. This would lead to a globalized and non-sectorized control.

## Figures and Tables

**Figure 1 pathogens-11-00668-f001:**
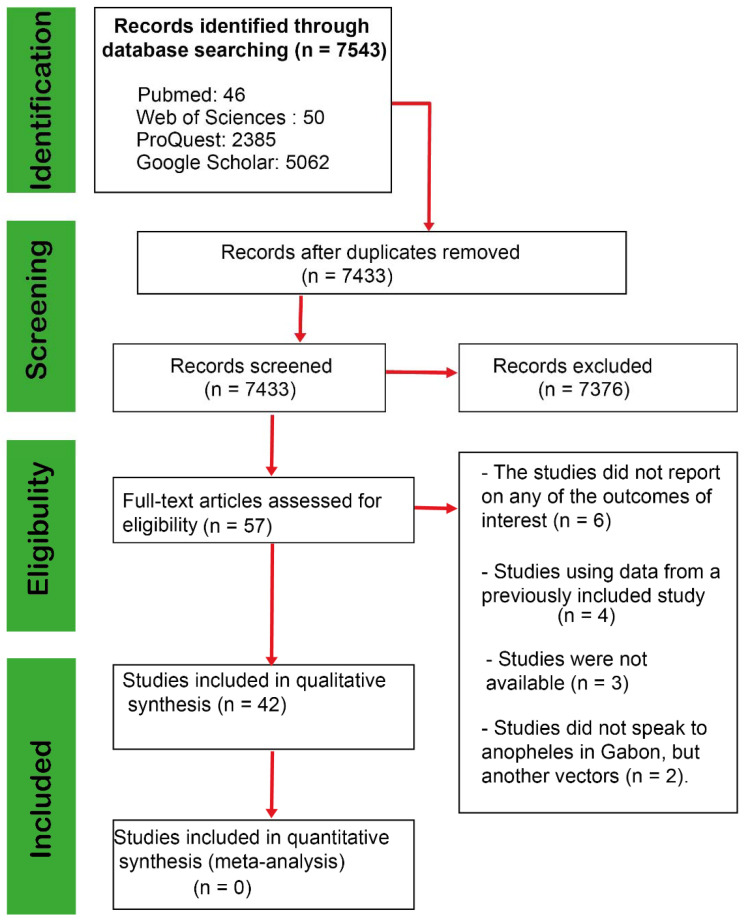
PRISMA flow diagram of search phases with numbers of studies included/excluded at each subsequent stage of the analysis.

**Figure 2 pathogens-11-00668-f002:**
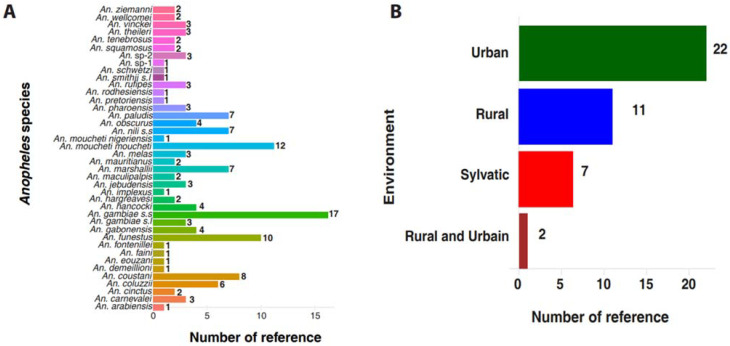
Characteristics of different studies included in this review. (**A**) Shows the number of studies that report the presence of a species. It shows that some species have been reported by at least two studies while only one has reported several species. (**B**) Shows the distribution of the studies carried out in each environment. It reveals that the studies concerned the following three different environments: urban, rural, and sylvatic.

**Figure 3 pathogens-11-00668-f003:**
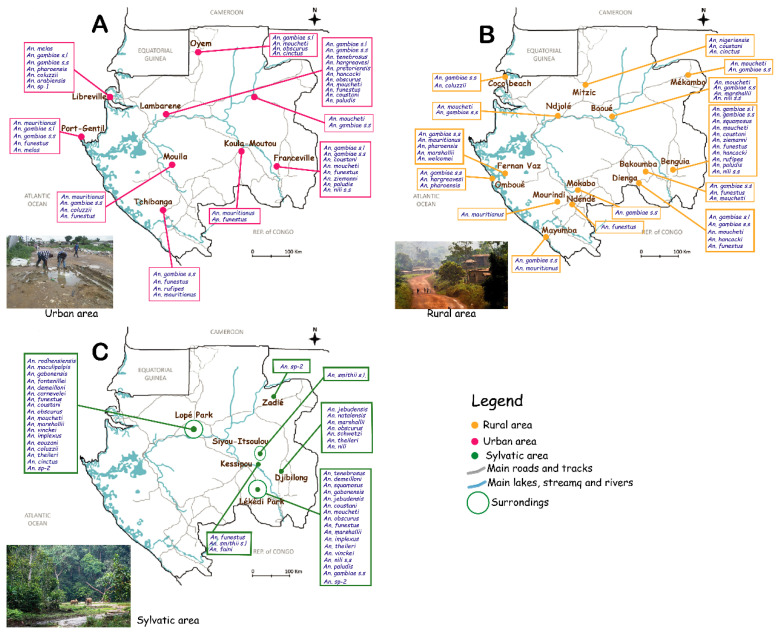
Diversity and spatial distribution of reported *Anopheles* species in Gabon. This map shows the different species of mosquitoes identified and recorded in Gabon in different habitats: (**A**) Urban, (**B**) rural, and (**C**) forest. The data contained in the map cover a period from 1931 to 2022.

**Figure 4 pathogens-11-00668-f004:**
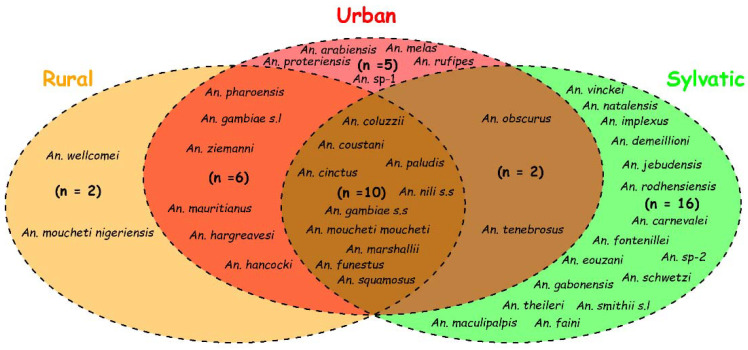
Diagram of relationship between different ecosystems and *Anopheles* species reported in Gabon between 1931 and 2022.

**Figure 5 pathogens-11-00668-f005:**
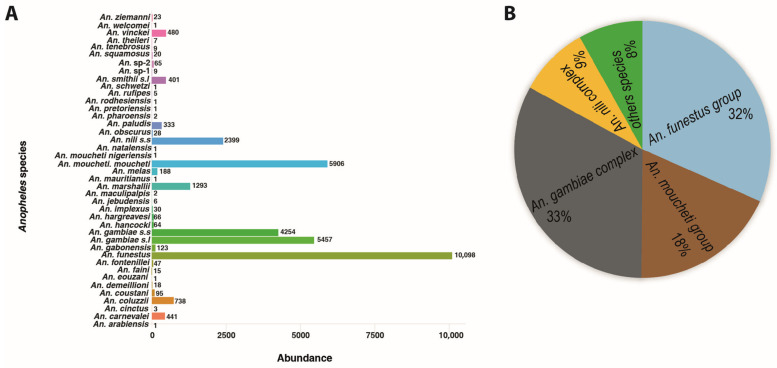
Cumulative abundance of each reported *Anopheles* species in different Gabon areas between 1931 and 2022. (**A**) presents the total number of individuals of each species reported. In contrast, (**B**) presents the frequency distribution of members of the Gambiae and Nili complexes and the Funestus and Moucheti groups.

**Table 1 pathogens-11-00668-t001:** Characteristics of *Anopheles* species reported in Gabon. *Anopheles* sp.-1 represents unidentified *Anopheles* specimen in urban areas and *Anopheles* sp.-2, unidentified *Anopheles* specimens from a forest environment.

Species	Abundance	Sites	Habitat	Vector Status	Reference
*Anopheles arabiensis*	1	Libreville	Urban	Major	[43]
*Anopheles carnevalei*	441	Lopé Park	Sylvatic	Minor	[25,32,33]
*Anopheles cinctus*	3	Lopé Park, Mitzic	Rural, Sylvatic, Urban	Minor	[25,42]
*Anopheles coluzzii*	738	Cocobeach, Libreville, Lopé Park, Mouila	Rural, Sylvatic, Urban	Major	[33,45,47,51,52,70]
*Anopheles coustani*	95	Benguia, Franceville, Lambaréné, Lékédi Park, Lopé Park, Mitzic	Rural, Sylvatic, Urban	Minor	[24,25,31,32,33,42,50,53]
*Anopheles demeillioni*	18	Lékédi Park, Lopé Park	Sylvatic	Undetermined	[25]
*Anopheles eouzani*	1	Lopé Park	Sylvatic	Undetermined	[33]
*Anopheles faini*	15	Kessipoughou Djibilong	Sylvatic	Undetermined	[71]
*Anopheles fontenillei*	47	Lopé Park	Sylvatic	Undetermined	[34]
*Anopheles funestus*	10,098	Benguia, Dienga, Eschiras, Franceville, Lambaréné, Lékédi Park, Lopé Park, Mouila, Ndendé, Port-Gentil, Tchibanga, Kessipoughou	Rural, Sylvatic, Urban	Major	[24,27,28,29,33,35,44,50,53,71]
*Anopheles gabonensis*	123	Lékédi Park, Lopé Park	Sylvatic	Undetermined	[25,32,33,49]
*Anopheles gambiae s.l.*	5457	Benguia, Franceville, Oyem	Rural, Urban	Major	[24,44,54]
*Anopheles gambiae s.s.*	4254	Cocobeach, Dienga, Fernan Vaz, Franceville, Lambaréné, Lékédi Park, Libreville, Mayumba, Mokabo, Mouila, Port-Gentil, Tchibanga	Rural, Sylvatic, Urban	Major	[25,26,27,28,29,31,35,44,45,47,50,51,52,53,55,56,68]
*Anopheles hancocki*	64	Benguia, Franceville, Lambaréné	Rural, Urban	Minor	[24,31,50,53]
*Anopheles hargreavesi*	66	Lambaréné	Rural, Urban	Undetermined	[30,31]
*Anopheles implexus*	30	Lékédi Park, Lopé Park	Sylvatic	Minor	[25]
*Anopheles jebudensis*	6	Lékédi Park, Djibilong	Sylvatic	Minor	[25,33,71]
*Anopheles maculipalpis*	2	Lopé Park	Sylvatic	Minor	[25,33]
*Anopheles marshallii*	1293	Fernan Vaz, Lambaréné, Lékédi Park, Lopé Park, Djibilong	Rural, Sylvatic, Urban	Minor	[25,28,29,31,32,33,71]
*Anopheles mauritianus*	7	Fernan Vaz, Mayumba, Mouila, Mourindi, Ndendé, Port-Gentil, Tchibanga	Rural, Urban	Undetermined	[28,29]
*Anopheles melas*	188	Libreville, Port-Gentil	Urban	Major	[26,47,53]
*Anopheles moucheti moucheti*	5906	Benguia, Dienga, Franceville, Lambaréné, Lékédi Park, Lopé Park	Rural, Sylvatic, Urban	Major	[24,25,27,31,32,33,35,44,48,50,53,56]
*Anopheles moucheti nigeriensis*	1	Mitzic	Rural	Minor	[42]
*Anopheles natalensis*	1	Djibilong	Sylvatic	Undetermined	[71]
*Anopheles nili s.s.*	2399	Benguia, Dienga, Franceville, Lékédi Park, Djibilong	Rural, Sylvatic, Urban	Major	[24,25,27,33,35,44,71]
*Anopheles obscurus*	28	Lambaréné, Lékédi Park, Lopé Park, Djibilong	Urban, Sylvatic	Minor	[25,31,32,71]
*Anopheles paludis*	333	Benguia, Franceville, Lambaréné, Lékédi Park	Rural, Sylvatic, Urban	Minor	[24,25,27,31,33,50,53]
*Anopheles pharoensis*	2	Fernan Vaz, Libreville	Rural, Urban	Minor	[28,29,66]
*Anopheles pretoriensis*	1	Lambaréné	Urban	Minor	[30]
*Anopheles rodhesiensis*	1	Lopé Park	Sylvatic	Minor	[33]
*Anopheles rufipes*	5	Franceville, Tchibanga	Urban	Minor	[24,28,29]
*Anopheles schwetzi*	1	Djibilong	Sylvatic	Undetermined	[71]
*Anopheles smithii s.l*	401	Kessipoughou Siyou Itsoulou	Sylvatic	Undetermined	[71]
*Anopheles* sp-1	9	Libreville	Urban	Undetermined	[57]
*Anopheles* sp-2	65	Lékédi Park, Lopé Park, Zadié	Sylvatic	Undetermined	[25,71]
*Anopheles squamosus*	20	Benguia, Franceville, Lékédi Park	Rural, Sylvatic, Urban	Minor	[24,25]
*Anopheles tenebrosus*	9	Lambaréné, Lékédi Park	Urban, Sylvatic	Minor	[25,31,33]
*Anopheles theileri*	14	Lékédi Park, Lopé Park, Djibilong	Sylvatic	Minor	[25,33,71]
*Anopheles vinckei*	480	Lékédi Park, Lopé Park	Sylvatic	Minor	[25,32,33]
*Anopheles wellcomei*	1	Fernan Vaz	Rural	Minor	[28,29]
*Anopheles ziemanni*	23	Benguia, Franceville	Rural, Urban	Minor	[24,27]

## Data Availability

Not applicable.

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
