# Peer review of "Systematic Review on Diversity and Distribution of *Anopheles* Species in Gabon: A Fresh Look at the Potential Malaria Vectors and Perspectives"

_pathogens, 2022, doi:10.3390/pathogens11060668_

Round 1
Reviewer 1 Report
This is an excellent review of the literature. It provides interesting knowledge on the diversity of anopheles in different ecosystems of the country. This article needs some modifications/explanations:
Abstract: lines 26, 27, 29 : please update the number of collected articles (7542), included studies (41) and species of anopheles (37).
Introduction : lines 58-61 According to WHO guidelines for malaria 2021, larviciding is considered as “a supplementary measure implemented alongside ITNs or IRS for urban areas where breeding sites are relatively few, fixed and findable in relation to houses”. Could you please modify the sentence to take into account these recommendations?
Results:
Please consider if this study met inclusion criteria as it add other anopheles species found in Gabon (An. smithii, An. faini, An. natalensis, An. schwetzi) : Obame-Nkoghe J, Rahola N, Ayala D, Yangari P, Jiolle D, Allene X, Bourgarel M, Maganga GD, Berthet N, Leroy EM, Paupy C (2017). Exploring the diversity of blood-sucking Diptera in caves of Central Africa. Scientific Reports, 7: 250. If yes, please add it and update the data and bibliography. If no, please detail the reasons for exclusion.
Line 152, Figure 2A, Table 1, Figure 5A: please correct the name of anopheles proteriensis to Anopheles pretoriensis
Figures 2B, 3 and 4: in order to match the text spelling, please replace selvatic by sylvatic
Figures 2A, 3, 5A: to follow the ordering of table 1, please sort the species names alphabetically
Table 1: for Anopheles funestus references: please change the semicolon in comma
Table 1: for Anopheles tenebrosus, please consider to add reference 25 and update the data
Figure 4: please update the number of Anopheles species (for example: rural only: n=2; sylvatic only: n=12)
Figure 5: please specify the total number of Anopheles
Conclusion:
Line 331: “there is no vector control strategy implemented in Gabon”: what about the strategy implemented by the “PNLP” with IRS and ITNs? This seems surprising; can you confirm this affirmation or mitigate this statement?
Conflicts of Interest: remove a point
Author Response
Reviewer 1
Abstract: lines 26, 27, 29: please update the number of collected articles (7542), included studies (41) and species of anopheles (37).
Thank you, we edited difference sentences (see lines 26, 27 and 29 in new manuscript version)
Introduction: lines 58-61 According to WHO guidelines for malaria 2021, larviciding is considered as “a supplementary measure implemented alongside ITNs or IRS for urban areas where breeding sites are relatively few, fixed and findable in relation to houses”. Could you please modify the sentence to take into account these recommendations?
According to your recommendations, the sentence has been changed (see lines 58-51).
Please consider if this study met inclusion criteria as it add other anopheles species found in Gabon (An. smithii, An. faini, An. natalensis, An. schwetzi): Obame-Nkoghe J, Rahola N, Ayala D, Yangari P, Jiolle D, Allene X, Bourgarel M, Maganga GD, Berthet N, Leroy EM, Paupy C (2017). Exploring the diversity of blood-sucking Diptera in caves of Central Africa. Scientific Reports, 7: 250. If yes, please add it and update the data and bibliography. If no, please detail the reasons for exclusion.
Thank you very much for your suggestion. This article meets the criteria for inclusion. Therefore, we have included this article in the study. The information contained in this article, such as Anopheles species and their abundance, has been included in all figures and tables in the article (for example see Fig 3 and Table 1).
Line 152, Figure 2A, Table 1, Figure 5A: please correct the name of Anopheles proteriensis to Anopheles pretoriensis
We changed “Anopheles proteriensis” to “Anopheles pretoriensis”(see for example line 152, Figure 2A, Table 1 and Figure 5A)
Figures 2B, 3 and 4: in order to match the text spelling, please replace selvatic by sylvatic
This has been done following referees’ recommandations, Figure 2B, Figure 3, and Figure 4
Figures 2A, 3, 5A: to follow the ordering of table 1, please sort the species names alphabetically
Thank you very much for your suggestion. Alphabetical order already exists except that it is from the origin of the 0 mark upwards.
Table 1: for Anopheles funestus references: please change the semicolon in comma
Thank you, we edit the Table 1
Table 1: for Anopheles tenebrosus, please consider to add reference 25 and update the data
Your suggestion has been taken into account and the data has been updated in Table 1, Figure 2 and Figure 5.
Figure 4: please update the number of Anopheles species (for example: rural only: n=2; sylvatic only: n=12)
Done, Figure 4
Figure 5: please specify the total number of Anopheles
The total number of mosquitoes of the genus Anopheles has been reported in the text in line 3 of chapter 3.5”Abundance of Anopheles species reported”
Conclusion: Line 331: “there is no vector control strategy implemented in Gabon”: what about the strategy implemented by the “PNLP” with IRS and ITNs? This seems surprising; can you confirm this affirmation or mitigate this statement?
Thank you for your comment. The sentence has been rearranged
Conflicts of Interest: remove a point:
we done
Reviewer 2 Report
I read with interest the manuscript number 1707034, titled: “Systematic review on diversity and distribution of Anopheles species in Gabon: a fresh look at the potential malaria vectors and perspectives”.
The review revealed and described the presence of several species of Anopheles collected in different locations in Gabon (including both sylvatic, rural and urban environments). The manuscript is very well written; clear, precise, and easy to understand.
Minor revisions:
Please, in the paper, convert Anopheles species in Anopheles species (lines: 24,25,28,29,35,38,141,167,169,171,180,183,185,201,211,317,334);
Line 359: delete one point;
Figures 2 and 5: Please, list the Anopheles species in alphabetical order (starting from An. arabiensis)
Author Response
Reviewer 2:
Please, in the paper, convert Anopheles species in Anopheles species (lines: 24,25,28,29,35,38,141,167,169,171,180,183,185,201,211,317,334);
We done see the new version of manuscript
Line 359: delete one point;
Okay, thank you, we done
Figures 2 and 5: Please, list the Anopheles species in alphabetical order (starting from An. arabiensis)
Thank you very much for your suggestion. Alphabetical order already exists except that it is from the origin of the 0 mark upwards.